# Lexical and Grammatical Errors in Developmentally Language Disordered and Typically Developed Children: The Impact of Age and Discourse Genre

**DOI:** 10.3390/children8121114

**Published:** 2021-12-02

**Authors:** Aleksandr N. Kornev, Ingrida Balčiūnienė

**Affiliations:** 1Department of Logopathology, Saint-Petersburg State Pediatric Medical University, 194100 Saint-Petersburg, Russia; ingrida.balciuniene@vdu.lt; 2Department of Lithuanian Studies, Vytautas Magnus University, 44248 Kaunas, Lithuania

**Keywords:** developmental language disorder, language errors, grammatical errors, lexical errors, derivational errors, preschool age

## Abstract

Persistent lexical and grammatical errors in children’s speech are usually recognized as the main evidence of language delay or language disorder. These errors are usually treated as a sign of a deficit in language competence. On the other hand, some studies have revealed the same kinds of grammatical errors in children with developmental language disorder (DLD) and in typically developed (TD) children. Quite often, DLD children use grammatical markers properly, but sometimes they do this erroneously. It has been suggested that the main area of the limitations in DLD children is language performance but not language competence. From the perspective of the resource deficit model, the error rate in DLD children should be influenced by the cognitive demands of utterance and text production. We presume that different genres of discourse demand a different number of cognitive resources and, thus, should differently impact the error rate in children’s speech production. To test our hypothesis, we carried out an error analysis of two corpora of child discourse. The first corpus contained longitudinal data of discourse (personal narratives, fictional stories, chats, and discussions) collected from 12 children at four age points (4 years 3 months., 4 years 8 months., 5 years 3 months., and 5 years 9 months. years). Another corpus contained discourse texts (fictional stories and discussions) collected in the framework of a cross-sectional study from 6-year-old TD and DLD children; the DLD children had language expression but not comprehension difficulties. A comparative analysis between different discourse genres evidenced that the genre of discourse and age of assessment impacted the error distribution in the DLD and TD children. Such variables as the lexical and morphological error rates were impacted the most significantly. The results of the two studies confirmed our hypothesis regarding the probabilistic nature of lexical and grammatical errors in both DLD and TD children and the relationship between a cognitive loading of the genre and the error rate.

## 1. Background

Language development is considered an essential part of mental development. Language is one of the main components of verbal reasoning as well as communicative and social interaction skills. Following the elegant expressions of [1], from the first months of life, a child tries to ‘become a native speaker’ and ‘to be a proficient speaker’. ‘To become a native speaker’ means to learn the mother tongue, i.e., to acquire numerous language units and multiple rules and combine them for producing utterances following appropriate standards of phonology, vocabulary, and grammar. During this time, a child accumulates and elaborates a rich complex of declarative and procedural knowledge. ‘To become a proficient speaker’ means to master many complex and flexible skills and strategies for producing cohesive and coherent discourse texts according to multiple social and cultural traditions a child faces within his/her community. Despite the differences among numeral theories of speech/language development [2,3,4,5,6,7], scientists generally agree that during the first several years of life, a child makes many errors in the phonological, lexical, and grammatical domains. Along the second and third years of life, some children tend to use many frozen phrases and, thus, make fewer errors (i.e., the so-called expressive style of language development; see [8,9]). On the contrary, other children prefer to construct their utterances according to language rules they have deduced and make many more errors (i.e., so-called referential style of language development) [8,9]. According to a comprehensive longitudinal study based on the natural observation of native Russian language acquisition [10], until the 7th year of life, typically developed (TD) children sometimes make grammatical or lexical errors. For example, by the age of 7 years, Russian speaking children usually have acquired noun morphology and select proper case forms (nominative, genitive, dative, accusative, instrumental or prepositional case) while speaking; however, Russian nouns fall under three declensions, and these are further divided into declension paradigms and their sub-types [11]. The extremely complex system of noun declension provokes children to sometimes select incorrect declension forms (although the case form is usually selected correctly) even at the 6th–7th year of life [9]. Besides incorrect declension forms, occasional omissions of functional words and production of neologisms may be also observed.

On the other hand, some errors made by children are obviously incidental and might be considered as slips of the tongue (SoT)—‘unintended, nonhabitual deviation(s) from a speech plan’ [12] (p. 284). The phenomena of speech errors and SoT are usually explained differently. Speech error is usually recognized as a consequence of the temporary incomplete language competence [13] or a result is the simplification processes in the programming of utterance inherent to children during language acquisition [14]. Other explanations highlight the developmentally caused cognitive resource immaturity, which, in young children, underlies the inability to sustain error-free speech production [15,16]. There is considerable evidence that problem-solving activity and implicit learning are essential parts of language development [17,18,19]. When a child is acquiring a native language, he/she tries to listen their caregivers and other communicative partners carefully to select the more informative grammatical, phonological, and semantical segments in the speech stream and try to attribute them to some extralinguistic events, objects, or features. Then, a child usually tries to repeat some piece of utterance and to control the concordance between the target word/phrase and his/her own production. Additionally, a child detects scaffolding from a child-directed speech containing positive or negative evidence [20,21]. This complex multimodal activity should be served by several cognitive abilities and skills, such as executive functions, selective auditory attention, working memory, self-monitoring, and strategy shifting [14,22]. Most of these cognitive resources are not completely mature until the 8th–11th year of life [23,24,25]. The younger the child is, the more pronounced these limitations are. Later on, the maturity of the cognitive resources grows gradually along the preschool and school years [14,26,27]. Though language competence and discourse skills develop rapidly, they remain incomplete until adolescence [28]. Thus, in other words, children execute a complex work of learning a native language in the conditions of quite limited cognitive resources [16]. In the majority of children populations, regular speech errors disappear before the 6th–7th year of life (although, in different languages, these milestones may differ) [29]. 

However, let us look at the other direction of speech and language development, i.e., discourse development. It is well known that discourse production is loaded by some additional cognitive activities that depend on discourse genres. 

Following [14], the development of discourse skills and cohesion devices ‘[…] involves two different but related problems: (1) the reorganization of stored linguistic representations so that they can form a system, and (2) the creation of a control process which constrains connected discourse as it is produced in real time’ [14] (p. 62). The control process managed by the self-monitoring subsystem plays a very important role in discourse development. As verbal communication strategies and linguistic devices, the discourse genres are determined by particular cultural traditions, and a child should learn them properly. Hence, employing problem-solving strategy [19,30,31], he/she acquires some procedural knowledge and elaborates a set of unique skills. For example, a child has to acquire procedural knowledge necessary for planning and producing narrative, descriptive and other genres’ passages while building coherent discourse, for using proper discourse markers, etc. A set of verbal communication skills includes, for example, skills of self-monitoring during narration or conversation, turn-taking skills necessary in a dialogue, pragmatic skills (e.g., manipulation with different registers of communication), and many other verbal behavior skills [32,33]. During this period, the abilities of speech behavior self-monitoring are essential to reach the social-communicative standards. The development of self-monitoring proficiency depends on age [14,34,35,36] and the maturity of cognitive resources [34]. In a few psycholinguistic studies of cognitive demands of different discourse genres, some evidence was obtained that, for example, a conversation is less demanding, while an expository discourse is more cognitively demanding than a narrative one [37]. For example, in narrative production, a child has to arrange the story structure and to verbalize intentions and goals, emotions, and other mental states held by characters [38,39,40]. When producing a spontaneous (unprepared) narrative, one has to develop a structure of events and produce oral discourse almost in parallel. This activity demands high resources. The basic skills that a child must acquire are the following: plan a logically well-organized semantic design, transform it into a relevant propositional structure, generate a verbal and syntactically cohesive text, and narrate it fluently to a listener. This activity is cognitively more demanding than a conversation by means of, for example, a dialog. High cognitive loading in the conditions of limited mental resources may result in a trade-off effect [17,41,42]. Suppose the child feels this conflict between the task demands and his/her available resources. In that case, it is possible to explore several strategies: (a) to simplify the lexical or syntactic structure of the utterances or (b) to reduce or simplify the content the child intends to express in the discourse text [43]. For example, it has been found that, in DLD children, the number of noun tokens per word within a story correlated negatively with a story structure, episode completeness, and CL/CU quotient; in TD children, correlations between the given measures were not evidenced. Moreover, in DLD children, visual story complexity significantly increased the percentage of noun tokens (which should be recognized as a negative characteristic for narratives) and the total number of grammatical errors. The more complex a visual story (according to the number of protagonists, actions, and semantically relevant features) was, the higher the noun percentage (during narration, verbs are used quite frequently (since they express actions/events), while nouns are more typical for descriptive texts. On the so-called ‘narrativity index’, see [44].) and the number of grammatical errors were. Presumably, a child sometimes does not feel a conflict between the task demands and the limitations in his/her language resources and, thus, try to verbalize too sophisticated content by too complex language structures. In such a case, a risk of incidence of speech errors his/her discourse may rise.

It is common knowledge that in some children, lexical and grammatical errors stay rather resistant despite the normal intelligence, hearing, and/or the absence of gross neurological disorders. Children cannot master sufficient language skills without the remedial treatment provided by a speech/language pathologist. This subpopulation is usually recognized as developmentally language-disordered (DLD) children (formerly known as specifically language-impaired—SLI—children) [45,46,47]. Different models of the DLD have been suggested. In some of them, it has been proposed that children with the DLD are not sensitive to main language markers and cannot process some language units (inflections, suffixes, prefixes, etc.) properly [48]. In these models, the DLD is treated as deviance of language development and its leading cause is defined as the domain-specific impairment. ‘For many years, this was seen as a consequence of a deficit either in perceptual processing or in underlying language representations, depending on one’s theoretical persuasion’ [49] (p. 5). However, in some studies, it has been found that the DLD children made the same errors as the younger TD children [50]. For example, some comparative studies of grammatical features between the DLD children and younger TD children with the same MLU rate did not reveal any significant distinctions [51].

Moreover, to review the basic lexical and grammatical errors in DLD children, most of them can also be found in TD children at the early stage of their language development [52]. Thus, DLD children are recognized as children with delayed language development [45]. Adherents of the given view believe that slow and improper language development in DLD children is not caused by the inability to learn language rules or low perception of particular linguistic features. Instead, it is supposed that the main limitations preventing these children from typical language development are limited cognitive resources domain-general models [52,53]. 

On the other hand, several studies in DLD children have revealed some weaknesses of the executive functions (EF), such as working memory deficit, low cognitive flexibility, selective inhibition deficit, and impulsivity [54,55]. The more complex the speech programming activity is, the more essential high EF resources are required [56,57]. During the utterance programming, different cognitive actions (lexemes and word form selection, serial order lexical arrangement, inflectional morphemes selection, etc.) compete for the cognitive resources [22,58].

As for speech/language errors in DLD children, it is usually implicated that their language drawbacks in discourse production are the same as in sentence production. However, some studies have revealed limitations in the discourse production only, which is not a consequence of the low language competence [59,60,61]. The majority of such limitations were revealed in story (re-)telling. DLD children produced less structured and cohesive texts with poor macrostructure and erroneous microstructure in comparison to their TD peers [43,62,63,64]. Among the different causes of the discourse drawbacks, scholars have highlighted the cognitive resource deficit found in DLD children [65]. It was found that many DLD children have weak EF [66] and a small volume of working memory [67]. In the DLD population, more severe and persistent EF deficiency prevents children from developing proficient language performance skills. Due to the deficit of EF, the error rate in DLD children discourse usually rises [45]. It was established that in discourse production, different genres have distinct procedural demands [68] and, thus, can provoke speech/language errors to different extents [69]. 

Several studies have revealed that the discourse genre significantly impacts the part-of-speech distribution both in the TD and in the DLD children [70]. Thus, it is reasonable to expect an impact of the discourse genre on the error rate in children’s speech. It was hypothesized that children’s errors while producing a discourse might have different domain-specific (linguistic limitations) and domain-general (cognitive resource deficit) mechanisms. The former should result in more regular errors, while the latter should result in more incident errors. It seems reasonable to expect speech error distribution to be a rather variable measure. Both in TD and DLD children, speech error number and distribution might be influenced by multiple variables, such as age, language competence, level of discourse skills, individual cognitive resource, genre of discourse, and/or register of communication. Thus, the quantitative statistical measures of the error distribution should be informative to understand the nature of errors.

The aim of the current research was to test the hypothesis that the cognitive loading of discourse may differently provoke lexical and grammatical errors in TD (at different stages of age) children and in DLD children (in comparison to TD peers). The prediction was that different biological age, as well as different registers of communication and the (sub-)type of discourse genres differently impact the distribution of lexical and grammatical errors. In study 1, we aimed at assessing the impact of age on the lexical and grammatical error distribution in different discourse genres in TD children. In study 2, we presumed to reveal differences between the impact of genre on the lexical and grammatical error distribution between the TD and DLD children. 

## 2. Materials and Methods

According to the main aim of the current project, two data sets were composed. Study 1 enabled us to assess longitudinally a group of TD children and to elicit discourse texts of different genres from the same children at four different age stages (Table 1 and Table 2). In study 2, two groups of children (DLD as the experimental and TD as the control group) were assessed cross-sectionally and the same discourse genres as in study 1 were elicited in slightly different semi-experimental conditions (Table 3 and Table 4).

### 2.1. Data Collection in the Study 1

#### 2.1.1. Participants

Participants were monolingual Russian-speaking children attending stated kindergartens in Saint-Petersburg, Russia (Table 1).

**Table 1 children-08-01114-t001:** Sample characteristics of study 1.

N	12			
Mean age	Wave 1	Wave2	Wave 3	Wave 4
	4 years 3 months	4 years 8 months	5 years 3 months	5 years 9 months
Language	Russian			
City of residence	Saint-Petersburg, Russia			
Language development	Typically developing			
Inclusion criteria	Normal non-verbal IQ			
Exclusion criteria	Hearing and/or visual disorders			
	Neurological disorders			
	Speech and/or language impairment			
	Non-verbal IQ on Raven’s matrix below 84			

In all participants, the nonverbal IQ (according to the Raven’s Colored Progressive Matrices Test) was at a normal range (*M* = 109.13, *SD* = 5.44).

Initially, 36 children were selected for the study from a large sample of 60 children previously screened by the speech-language pathologist and confirmed as as typically developed children without any speech/language disorder. The 36 children passed as many as possible of the designed assessment sessions within each of the waves; however, a few of the children were excluded from the study after the 1st, 2nd, or 3rd wave due to different reasons; moreover, not all the children participated in all the assessment sessions. Thus, finally, for the current study, we selected 12 children of the most similar age, who passed all the waves and all the assessment sessions within each of the waves.

#### 2.1.2. Procedure

The data contained genres such as personal narrative, fictional story (telling and retelling mode), chat, and discussion, in each wave of the assessment (Table 2). The given genres were different from the perspective of form (monologue vs. dialogue) and register of communication (peer-directed vs. adult-directed speech). The size (the total number of words) of the data is presented in Table 2.

**Table 2 children-08-01114-t002:** The data of study 1. The total number of words.

Genre	Register	Wave 1	Wave 2	Wave 3	Wave 4
Personal narratives	Peer-directed speech	1526	1869	3030	3164
Fictional stories (telling)	Peer-directed speech	544	484	706	743
Fictional stories (telling)	Adult-directed speech	576	541	557	868
Fictional stories (retelling)	Adult-directed speech	593	674	618	794
Chats	Peer-directed speech	1356	2406	3910	3216
Discussions	Peer-directed speech	1332	1670	1932	1859

The conversational map methodology [38] was modified by the authors of the paper to elicit peer-directed chats and personal narratives. During individual sessions, a child was given a doll (whose name was ‘the same as the child’s name’ and whose age was also ‘the same as the child’s age’), and then he/she was involved in a chat with another doll (the experimenter’s doll). In the flow of conversation, the experimenter’s doll told a personal narrative about some of his/her experience and then asked the child’s doll whether she/he has had a similar experience. In response to the given prompts, the children usually told personal narratives about similar experiences (events that had happened with the child or his parents/grandparents/siblings/friends, etc.). While a child was telling his/her narrative, the experimenter tried to minimize her impact, i.e., she did not help the child, did not ask any questions, and did not make any corrections. The only allowed prompts were neutral remarks, such as ‘Uh-huh’, or a repetition of the exact previous words of the child. In each of the waves, three personal narratives were elicited from each child. The experimenter introduced the following topics in the prompt narratives: shopping in a supermarket, visiting a doctor, a journey to a zoo, swimming in a lake, and some others familiar for children of the given culture and lifestyle. Moreover, the children were allowed to tell personal narratives on any topic they offered. 

To elicit adult-directed fictional stories, the children were asked to tell and retell a story according to different picture sequences. In each of the waves, we used two picture sequences consisting of six colored pictures each; the protagonists of the picture sequences were animals quite familiar for children of the given culture and lifestyle (e.g., cats, dogs, mice, crawls) but one of the stories was always more complex than another one. The pictures were designed by the authors of the paper and painted by a professional artist, Ms. Unė Kurtinaitytė. During the assessment, the experimenter first placed the pictures in the correct sequence in a single, horizontal row in front of a child. Then, the child was allowed to look at the pictures to get the gist of the story (the time was unlimited). Next, the child was asked to tell a story according to the pictures (for the telling mode) or to listen to the story read by the experimenter and then to retell it (for the retelling mode). During the telling/retelling process, the pictures were still on the table and, thus, a child had a possibility to look at them all the time of assessment. The order of the tasks was counterbalanced regarding story complexity (easier story vs. more complex story) and task mode (telling vs. retelling). Sessions of the 1st and 2nd tasks were separated by a few minutes of chat between the experimenter and a child. 

To elicit peer-directed fictional stories and discussions, we used the same dolls for stimulating the personal narratives and chats. At the beginning of the assessment, the experimenter’s doll demonstrated hand-made his/her ‘favorite book with pictures inside’ and offered the child to look at them. Then, the experimenter’s doll ‘accidentally‘ dropped the book, and the images spilled on the table (the images were not stuck initially to the pages of the book). The experimenter’s doll asked the child’s doll for help and they both tried to place the pictures to repair the picture sequence. During this activity, the experimenter’s doll offered erroneous versions of the picture sequence, asked many provoking questions, and, thus, involved the child in a discussion. 

### 2.2. Data Collection in the Study 2

#### 2.2.1. Participants

Participants were 10 monolingual Russian-speaking children with DLD and 14 TD peers (see Table 3).

**Table 3 children-08-01114-t003:** Sample characteristics of study 2.

	Experimental Group (DLD)	Control Group (TD)
N	10	14
Mean age	6 years 5 months	
Language	Russian	
City of residence	Saint-Petersburg, Russia	
Language development	Primarily impaired	Typically developing
Inclusion criteria	Normal non-verbal IQ	Normal non-verbal IQ
	Clinically referred DLD	
Exclusion criteria	Language comprehension disorders	Speech and/or language impairment
	Hearing and/or visual disorders	
	Neurological disorders	
	Non-verbal IQ on Raven’s matrix below 84	

DLD children were clinically referred and received a two-year course of speech therapy (five sessions a week) at the kindergarten; nevertheless, various phonetic, lexical, and grammatical errors still occurred in their speech data. Before the experiment, all the children were assessed by a speech-language pathologist by means of Russian language assessment tools [50] in order to confirm the TD vs. DLD status and to exclude children with language comprehension disorders from the experiment. (Children with language comprehension disorders were excluded to escape the non-relevant variable impact and to highlight the discourse production features related to the expressive language limitations.) Thus, the experimental group in our study may be characterized as DLD children with language expression but not comprehension difficulties. Nonverbal IQ (according to Raven’s Colored Progressive Matrices Test) was at a normal range in both groups (*M =* 113.25, *SD* = 3.15 in the DLD; *M =* 121.00, *SD* = 4.90 in the TD).

#### 2.2.2. Procedure

The data contained genres such as fictional story (telling and retelling) and discussion (Table 4). The given genres were different from the perspective of the form (monologue vs. dialogue) but belonged to the same register of communication (adult-directed speech). The size (the total number of words) of the data is presented in Table 4.

**Table 4 children-08-01114-t004:** The data of study 2. The total number of words.

Genre	Register	DLD	TD
Fictional stories (telling)	Adult-directed speech	520	1130
Fictional stories (retelling)	Adult-directed speech	314	1156
Discussions	Adult-directed speech	6249	8687

The children were asked to tell and retell a story according to different picture sequences to elicit fictional narratives. The order of the tasks was counterbalanced regarding story complexity (as in study 1, one of the stories was easier, while another one was more complex) and task mode (storytelling vs. retelling). The methodology has been previously presented in several publications [43,71].

To elicit discussions, the ‘nonsense picture method’ [72] was employed. An experimenter demonstrated a child a picture with an unrealistic scenario (e.g., a cow sitting on a tree, a pig flying in the sky, etc.) and asked the child to evaluate its plausibility. During the conversation, the experimenter asked the child as many as possible provoking questions and tried to involve the child in a discussion [72].

### 2.3. Development of the Corpus of Russian Children’s Language 

The data (audio-records of the sessions) were transcribed orthographically using CHAT tools [73]. Two experts double-checked the transcriptions independently and extended them by encoding for language errors and linguistic dysfluencies to perform automated analysis using CLAN tools [73]. The data analyzed comprised, in total, 45, 942 words without mazes.

### 2.4. The Analyzed Variables

Russian is a highly inflective and morphologically rich language with many grammatical categories. First, Russian nouns are specified as animate vs. inanimate. Furthermore, as mentioned above, Russian nouns fall into three declensions related to the gender form and may be further divided into different declension paradigms; moreover, nouns involve two number forms and six case forms. Each Russian verb is specified for perfective vs. imperfective aspect; the given forms of aspect are differentiated by prefixation, suffixation (or a combination of them), and root alternation (suppletion). Verbs are also specified for reflexive vs. non-reflexive form, they involve three tense forms (past, present, and future tense) and three mood forms (indicative, imperative, and conditional). There are five participles different in passive vs. active form and in perfective vs. imperfective aspect. Participles are declined as adjectives; the two indeclinable adverbial participles are often called gerunds. In Russian, nouns agree with their modifiers in gender, case, and number; verbs agree with nouns and pronouns in gender, person, and number. Word order in Russian is syntactically flexible and determined by pragmatics [11]. The most common syntactic structure ‘subject-predicate-object may have up to 30 possible word order variants, including six with the subject omission [74]. In addition, subordination using participles and gerunds instead of relative and adverbial clauses is common in Russian.

A significant number of studies in Russian child language has been devoted to grammar acquisition [10,75,76], and, thus, the qualitative features of morphological errors in the early noun, verb, pronoun, and adjective production are well-studied; however, quantitative studies have been carried out much rarely, especially at the later stages of language acquisition. Similarly, the qualitative features of lexical and derivational errors have been widely analyzed [77,78], however, we still lack knowledge about their quantitative features. Therefore, this study analyzed both qualitative and quantitative features of the following types of language errors, as exemplified below. 

Lexical errors were considered as cases in which the child referred to something by another name. In our corpus, lexical errors fell into (a) hyponyms replaced by hyperonyms (e.g., ‘mouse’s house’ instead of ‘mouse’s hole’), (b) hyperonyms replaced by hyponyms (e.g., ‘chicken’ instead of ‘baby-bird’), and (c) incorrect selections within words of the same semantic group (e.g., motion terms ‘to go’, ‘to run’, and ‘to fly’).

Morphological errors were considered as cases in which the child selected an inappropriate form for marking the case, declension paradigm, gender, number, person, tense, aspect, mood, or other morphological categories. Typical examples of morphological errors were as in the following:

(1) *Tam byla* [= byl] *poni.* ‘There was:FEM [= was: MASC] a ponny: MASC.’

(2) *Koshka zabiralasj* [= zabralasj] *na derevo.* ‘The cat was climbing [= climbed] up the tree.’

In the case (1), the child used a feminine form for the word ‘was’ instead of a masculine form, while the noun ‘ponny’ falls to the masculine gender. In the case (2), the child used incorrectly an imperfect aspect form instead of the proper perfect form.

Syntactic errors were divided into (a) the agreement errors, (b) the government errors, (c) the incorrect prepositional constructions, (d) the omissions of functional words, (e) the incorrect word order, and (f) other ’messy sentences’. Typical examples of the syntactic errors were as in the following:

(3) *Ona stala zalezatj k derevu* [= na derevo]. ‘She started climbing to [= up] the tree.’(4) *Ptica ne mozhet letatj s vodoj* [= v vode]. ‘A bird cannot fly with [= in] a water.’

In both cases (3,4), the child used incorrect preposition (although the government of the preposition on the noun case form was correct). The given syntactic errors (3,4) not only destructed the sentence but also involved inappropriate semantics.

Derivational errors fell into (a) the incorrect prefixation, (b) the incorrect suffixation, and (c) the incorrect compounding. Usually, derivational errors were occasional word-forms constructed by a child (e.g., a diminutive form of ‘a pet’ or ‘a swing’ that are possible theoretically but do not exist in Russian). 

During statistical analysis, the quotient for each type of error was estimated by dividing the number of errors of the given type by the total number of words in the corpus. Thus, four different quotients were estimated: (1) the lexical error Q_L_, (2) morphological error Q_M_, (3) syntactic error Q_S_, and (4) derivational error Q_D_. The statistical difference between the measures was calculated by means of the ANOVA (in the case of normally distributed variables) or by the Mann–Whitney U criterion (in the case of non-normally distributed variables). The MANOVA (general linear model—GLM) was applied to estimate the determinants’ impact on the dependent variables. 

## 3. Results

In this section, we present separate results of study 1 and study 2.

### 3.1. Results of the Study 1

Although the participants had no speech/language disorders, they made a low number of errors, i.e., 0.02 errors per word (or 20 errors per 1000 words) with high individual variability (from 0 to 0.19). As for different types of errors, their distribution in the whole data was uneven (Figure 1).

Lexical (Q_L_) and morphological (Q_M_) error quotients were significantly highest among all types of errors, while syntactic (Q_S_) error quotient was the lowest (Table 5).

Along the waves, the mean of the total error quotient (Q_T_) changed following the U curve, but the differences between the waves did not reach a significant level. The maximal error rates were evidenced in the 1st wave, while the minimal error rates were evidenced in the 3rd wave (Figure 2).

Distribution of the quotient for all types of errors are presented in Figure 3.

The total number of errors per word changed gradually from wave to wave but all types of errors had their specific patterns. The most dramatic change was revealed between the 2nd and the 3rd wave.

A more selective quantitative analysis of the rate of different types of errors in each wave revealed distinctions in the errors’ ‘behavior’. The lexical error rate was maximal in the 1st wave and minimal in the 3rd wave (U-shaped pattern), while the syntactic error rate, vice versa, was minimal in the 1st and in the 4th wave and maximal in the 3rd wave (inverted U-shaped pattern), and the morphological errors reduced gradually along the waves.

As for the percentage of different types of errors among all errors, it was rather stable in the first three waves but changed in the 4th wave (Figure 4).

To estimate the impact of such determinants as Wave and Genre on the error distribution, the MANOVA analysis of the total data in all the waves was carried out (Table 6).

Both Wave and Genre significantly impacted the Q_L_ and Q_M_ indexes with relatively low effect size.

Comparative between-genre ANOVA analysis with a post hoc multiple pairwise comparisons revealed that the Q_L_ quotient in the storytelling was higher than in the personal narrative (*p* = 0.001), retelling (*p* = 0.022), chat (*p* = 0.000), and discussion (*p* = 0.000). As can be seen in Figure 5, the register of communication (adult-directed story vs. peer-directed story) significantly influenced only the distribution of lexical errors in the storytelling.

### 3.2. Discussion on the Study 1

The aim of study 1 was to assess the impact of Age variable on the lexical and grammatical error distribution in different discourse genres in TD children. Distributional analysis of the total number of errors revealed that the lexical and morphological errors were dominant, while the syntactic and derivational ones were much rarer. This result is congruent with our knowledge about the patterns of development of different parts of language devices [11,78,79]. The basic syntactic skills become matured in the second and third years of life. Acquisition of the derivative words and mastering the derivational morphemes begin at about 2 years 6 moths and develop the most active during the 4th and 5th years, usually resulting in a high number of word innovations. On the other hand, the differentiation between the word meanings and the development of a polysemy continue along the school years until adolescence. Basic morphology devices are mastered by typically developed children at about the 6th or 7th year of life. In this regard, we can infer that earlier developed speech/language skills are more resistant to errors, while later developed ones, conversely, are more variable and more error prone. Our participants’ total number of errors significantly reduced in the 3rd wave compared to the 1st one. Until recently, such data regarding the changes in error patterns during late preschool years were not available. A small amount of data with error samples have been obtained instead in special experimental assessments, but not in the spontaneous discourse; spontaneous data, if accepted, have not been supported by an age-related analysis [79,80]. To our knowledge, there have not been any previous attempts to analyze the entire range of error profiles in TD children’s discourse.

It is difficult to compare our data with other data. Among the publications related to grammatical errors in children’s speech [80,81], very few contain statistics of distribution. For example, [80] presented a collection of child errors, and in his corpus, lexical errors occupy 92% and morphological 8% of all errors. This is much less than in our data (35%). Furthermore, it should be noted that in different languages the weight of the formally similar types of errors may have a different mechanism and may be grounded in different strategies of the speaker.

Analysis of the age-related changes in four types of errors presented new data about the discourse development in children from 4 to 6. As shown in Figure 2 and Figure 3, participants became more skillful in the discourse production during the two years of a longitudinal study. The total number of errors changed following the U-shaped curve. This trend is relevant to many other developmental studies representing the same pattern [2,82]. However, the rate of different types of error changed according to different patterns. The most clear-cut patterns of changes were presented in the lexical and morphological errors. The morphological errors gradually reduced and followed a descending linear curve. On the other hand, the lexical errors path was observed as a U-shaped curve—they radically reduced after the 2nd wave (mean age 4 years 8 months) and raised after the 3rd (mean age 5 years 3 months). One more interesting point in our data was the internal distribution of different types of errors. The pattern of this distribution was constant along the first three waves and changed in the 4th wave. This occurred because the rate of the morphological errors was raised, but the lexical errors were reduced in only the 4th wave. Concurrently, the remaining two types of errors remained constant. This could probably be explained by the fact that age from 5 to 6 is critical for the development of some cognitive resources and executive functions [26,83,84]. 

In our participants, individual variabilities of error distribution were probably caused by the two main circumstances: (a) partial immaturity of some discourse language skill subsystems and (b) different cognitive demands of the discourse genres. The last inference is supported by our finding of the distinct and significant impact of different genres on the error distribution. Moreover, in different types of error this impact manifested with distinct patterns (Figure 5).

To our knowledge, this study is the first to confirm the impact of age and genre on the error distribution in the child discourse. Moreover, statistical analysis revealed that the assessed four types of errors were differently sensitive to age and genre. It should be emphasized that the four types of errors represent the basic sub-systems of language development and distinct patterns of language maturity: (1) growth of lexical and conceptual diversity, (2) development of inflectional morphology devices, (3) development of syntactic structures, and (4) development of derivational morphology devices [79]. Only the lexical errors were similarly by the genre and only the morphological errors were sensitive to age. Some well-known developmental patterns can explain this. According to [10,78], children continue to master morphology devices for noun inflections which play a crucial role in Russian. However, this grammatical device seems to be not fully mastered and not yet automatized in most of our participants. As for the accuracy in the vocabulary use, children of this age range acquire many new infrequent words relatively fast, but the semantic differentiation develops later [79]. Children’s word choice errors reflect incomplete knowledge of the meaning of the incorrectly used words [2]. The discourse programming demands more conceptually rich vocabulary than utterance production. In this regard, lexical errors are multiple and influenced by age and genre. The same data has been published previously by other authors [2,79].

### 3.3. Results of the Study 2

The aim of the study 2 was to reveal differences between the impact of genre on the lexical and grammatical error distribution between the TD and DLD children. The total number of errors per word in the DLD group was higher than in the TD group (*F* = 6.114; *p* = 0.025) (but only in the conversation). Comparative estimation of different types of errors revealed that the DLD children made significantly more lexical errors per word (*F* = 4.530; *p* = 0.037) and almost significantly more syntactic errors (*F* = 3.719; *p* = 0.058). 

The MANOVA analysis of the impact of Group and Genre on the error rate in all participants (DLD + TD) revealed that both Group and Genre determinants predicted the lexical and morphological errors rates (Table 7).

Group significantly determined the morphological errors and almost significantly determined the lexical and syntactic errors. On the other hand, Genre significantly determined the lexical errors. The comparative between-group analysis of error distribution in the storytelling and retelling did not reveal significant distinctions between the groups. However, the same analysis of error distribution in the discussion revealed different results: the DLD children made significantly more lexical errors (*U* = 11.5; *p* = 0.018) and syntactic errors (*U* = 13.5; *p* = 0.032) (Figure 6).

To estimate the Genre’s impact on the different errors’ rates within the same group, a comparative analysis of errors per word in the storytelling and conversational reasoning was carried out. In the DLD group, we did not find a difference between the genres. In the TD group, significant differences between the storytelling and conversation were in the morphological (*U* = 34.0; *p* = 0.000) and syntactic errors (*U* = 70.0; *p* = 0.034) (Figure 7).

Statistical comparison (Mann–Whitney test) of the error quotient rate distribution in storytelling, retelling, and discussion evidenced that Genre (storytelling vs. discussion) significantly discriminated the Q_M_. (*U* = 119.0; *p* = 0.006), Q_S_ (*U* = 150.0; *p* = 0.013), and Q_D_ (*U* = 139.5; *p* = 0.006) (Figure 7).

Comparative analysis of the distribution of error quotient in different genres in the DLD group confirmed that Genre (storytelling vs. discussion) significantly discriminated only the syntactic errors (*U* = 22.5; *p* = 0.006). However, a comparison of the percentage of the types of errors among all errors in the DLD children in different genres revealed significant differences between the lexical (*U* = 82.0; *p* = 0.010) and syntactical errors (*U* = 16.5; *p* = 0.017).

### 3.4. Discussion on the Study 2

At the beginning of the discussion, we need to remind that all DLD participants were attending special kindergartens for DLD children and receiving remedial treatment course. At the beginning of the experiment, the DLD participants had been taking part in the remedial treatment for approximately two years. As a result, some speech/language drawbacks (especially, in morphological and syntactic domains) had probably been partially compensated before the beginning of the experiment. Nevertheless, the scores of language assessment (phonological and morphosyntactic tasks) used to select the children for the experiment were below the age-norms in all the DLD children. Moreover, it was noticed that in the semi-structured discourse elicitation conditions, many of the DLD participants were trying to escape linguistically demanding phrases. Comparative analysis of narrative microstructure in the same sample of the DLD and TD children (see [43]) revealed that the DLD children tended to use simple syntactical structures, their communication units were not complex (the mean number of clauses per communication unit was low), and the lexical diversity index (noun lemma/token ratio) was also quite low. Although the DLD and TD children did not demonstrate significant differences in the total number of errors in storytelling, discussions about the ‘nonsense pictures’ provoked the DLD children to make more errors than the TD children. This result concord to studies carried out in other languages [85] and might be explained by different nature of the given genres. Narrative (storytelling according to picture sequence in a self-pace mode) might be characterized as partially prepared and structured speech, while the discussion is much more spontaneous and much less structured genre. Moreover, in the current study, Genre impacted differently not only the total number but also a distribution of different types of errors. Namely, cognitively high demanding tasks (such as storytelling elicited following quite structured procedure) provoked the DLD children to make a lot of morphological errors; while spontaneous, playful discussions with an experimenter lead the DLD children to numerous lexical errors. This tendency concords with our previous experiments [71]. Thus, the results of study 2 evidenced a selective impact of the discourse genre on error distribution in the DLD children.

### 3.5. Results of Comparison between the Study 1 and Study 2

Comparison of the error rates between the TD children from the study 1 wave 4 (mean age 5 years 9 months), the TD children from study 2 (mean age 6 years 5 months), and their DLD peers (mean age 6 years 5 months) did not reveal any significant distinctions (Figure 8).

## 4. General Discussion on the Study 1 and Study 2

Our two studies evidenced that the correctness of speech production in the same children is not a constant measure. The high variability of error scores was significantly influenced by the genre of discourse and the age of participants. Thus, our prediction about the impact of the genre on the error rate was confirmed. This may be explained by the different cognitive loading of different genres. These data are congruent with our previous studies where the genre impact on the distribution of phonological structures (types of syllables with different complexity) [71] and on the part-of-speech profile [70] was evidenced.

It should be noted that the distribution of the percentage of different types of errors was very similar between the TD and DLD children. However, the latter made significantly more lexical errors.

According to the traditional attitudes of speech/language pathology, low language competence is the main distinction between language disorders and TD children. The development of language competence is usually equated to the score of correct responses in language battery tasks (the proper response is considered the response without errors). Thus, it is implicitly suggested that typically developed children do not make any errors. However, this suggestion is not true. Both adults and children make errors and slips of the tongue in their colloquial speech [2,86]. While there are plenty of psycholinguistic studies of errors in colloquial speech in adults, scarce data are available regarding colloquial speech and errors in children. From the psycholinguistic and psychological perspective, speech is some kind of behavior consisting of multiple skills. From this perspective, procedural knowledge acquisition in speech development is in close relation to mastering relevant skills. 

In linguistic studies of children’s language, speech errors are recognized as a sign of limited language competence. However, language competence is usually rather stable, and its limitations are expected to manifest approximately similar to different speech acts.

Like many other skills, each new speech skill passes through a stage of unstable, variable execution, and automation. The less mature and automated the skill is, the more variable it is, and the more mistakes are made [87,88]. The number of errors the subject makes is a manifestation of the skill’s instability. From the perspective of this regularity, speech errors mark the weak chains in a particular speech/language subsystem. In the scope of this suggested model, higher rates of lexical and morphological errors, compared to syntactic and derivational errors, mean that the syntactic and derivational subsystems, in 4–5-year-old children, are more matured and automated and, therefore, children make fewer errors. For the same reason, the syntactic and derivational errors were less sensitive to age and genre.

The main limitations of our study were relatively small size of the sample and quite narrow range of cognitive assessments. In the framework of the ongoing investigation, we are replicating the longitudinal study (with an application of all methods and approaches employed in the given study 2) in DLD children. This will enable us to compare a process of discourse acquisition between the TD and DLD children from different perspectives, including error analysis.

## 5. Conclusions

Comparison of the data obtained in the longitudinal and cross-sectional studies evidenced that both in TD and DLD children, genre of discourse significantly impacted the distribution of lexical and grammatical errors. Different genres had a different impact on the pattern of the error distribution. On the other hand, different types of errors were influenced by the genre to different extents. Lexical errors were the most sensitive to the genre both in TD and DLD children, but the distribution of only morphological errors discriminated TD children from DLD peers. Moreover, the general patterns of error distribution in different genres of discourse are reasonably similar between TD and DLD children. 

Our data confirmed that methods, tasks, and approaches to children’s speech/language assessment may have an essential impact on the error rate. Thus professionals (e.g., speech/language pathologists) should consider the genre and communication register. According to our current data, individual variabilities of different types of grammatical devices are not equally sensitive to the demands of different genres.

## Figures and Tables

**Figure 1 children-08-01114-f001:**
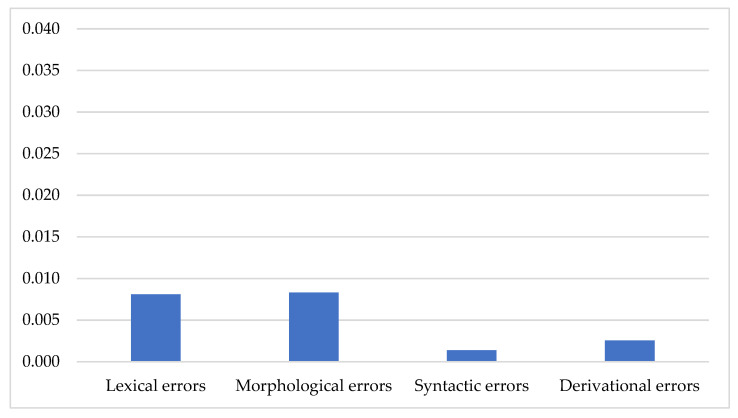
Error rate quotients per word in different types of errors.

**Figure 2 children-08-01114-f002:**
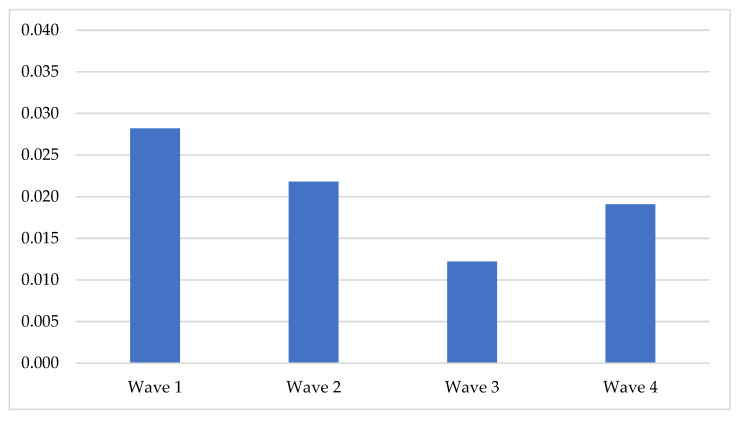
Total error rate per word in the waves of assessment.

**Figure 3 children-08-01114-f003:**
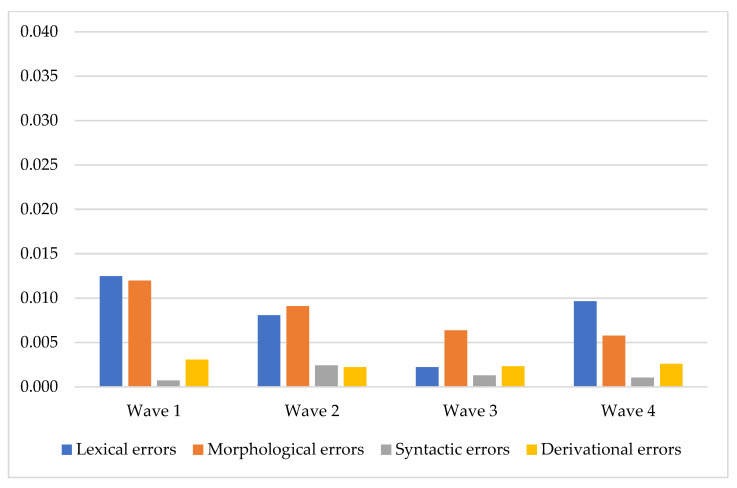
Distribution of different types of error quotients in the waves of assessment.

**Figure 4 children-08-01114-f004:**
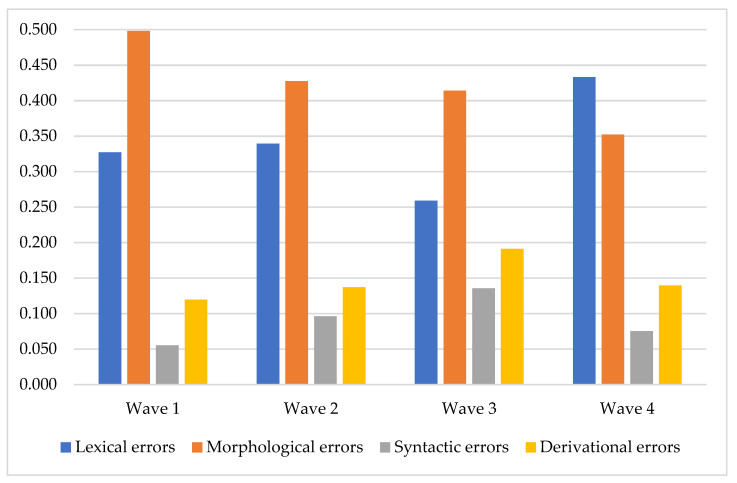
The percentage of different types of errors (among all errors) in the waves of assessment.

**Figure 5 children-08-01114-f005:**
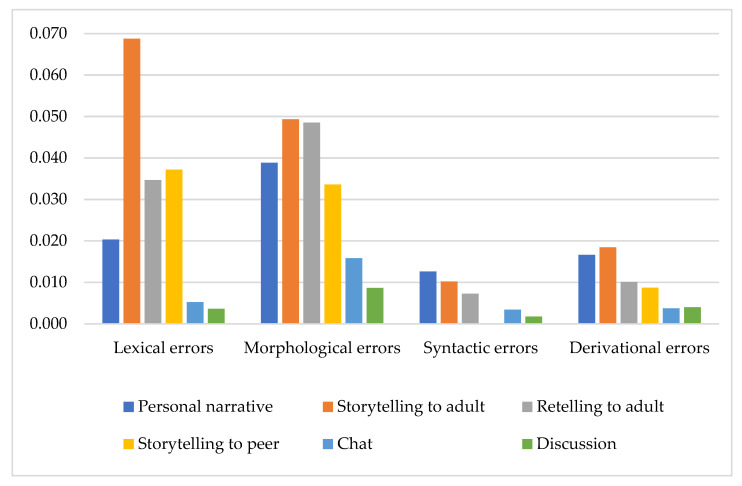
The impact of *genre* on error distribution.

**Figure 6 children-08-01114-f006:**
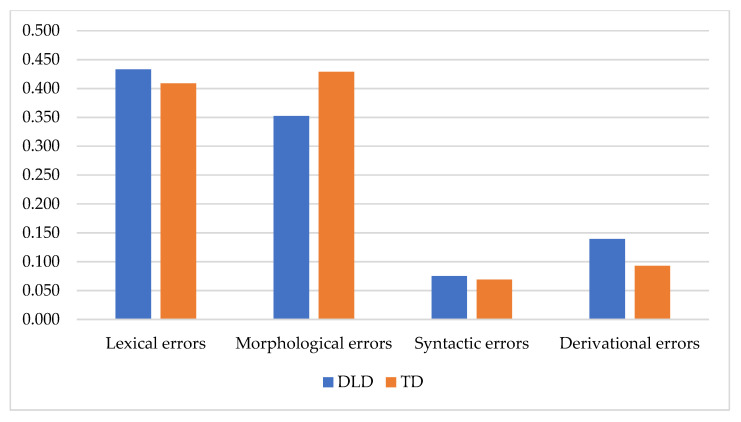
Error percentage distribution in TD and DLD children.

**Figure 7 children-08-01114-f007:**
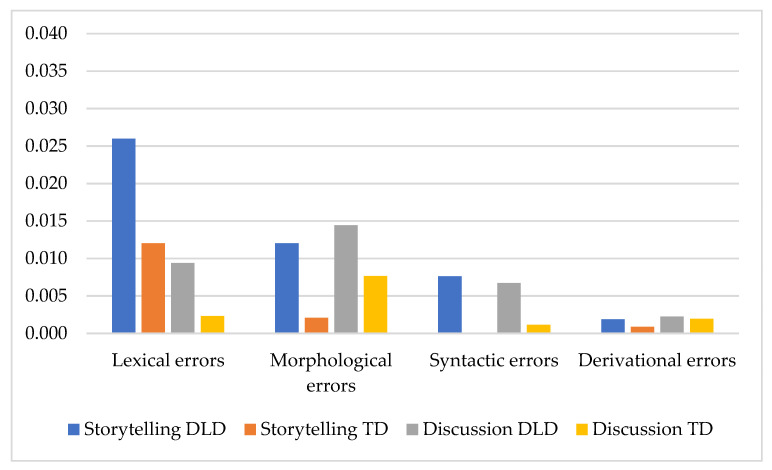
Error distribution in the storytelling and discussion. The number of errors per word.

**Figure 8 children-08-01114-f008:**
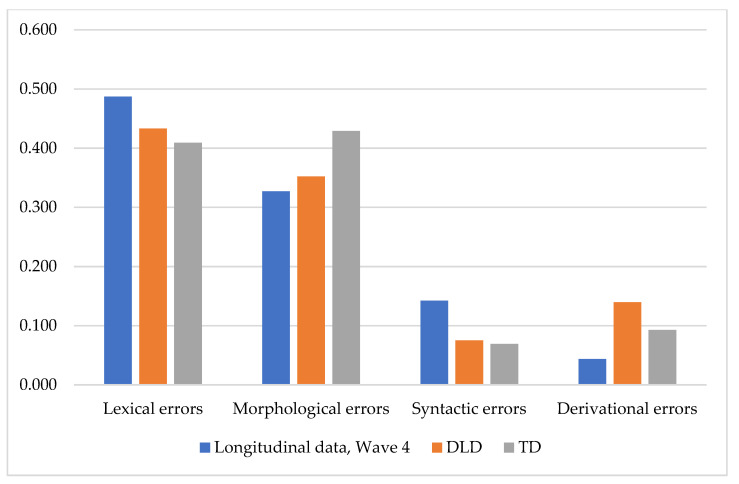
Distribution of the percentage of different error types within the groups from study 1 and study 2.

**Table 5 children-08-01114-t005:** The error quotient among the waves.

	The Number of Errors Per Word Mean (Standard Deviation)			
	Lexical Errors	Morphological Errors	Syntactic Errors	Derivational Errors
Wave 1	0.0125 (0.0270)	0.0173 (0.0019)	0.0007 (0.0030)	0.0031 (0.0088)
Wave 2	0.0081 (0.0189)	0.0153 (0.0017)	0.0024 (0.0082)	0.0022 (0.0077)
Wave 3	0.0022 (0.0048)	0.0118 (0.0013)	0.0013 (0.0037)	0.0023 (0.0069)
Wave 4	0.0096 (0.0143)	0.0076 (0.0008)	0.0011 (0.0032)	0.0026 (0.0057)
Significance				
1–2	0.690	1.000	0.178	1.000
1–3	0.002	0.048	1.000	1.000
1–4		0.021	1.000	1.000

**Table 6 children-08-01114-t006:** The impact of group and genre determinants on error distribution.

Dependent Variables	Independent Variables			
	*F*	*p*	*η^2^*	*Power*
	Wave			
Lexical errors per word	5.365	0.001	0.050	0.932
Morphological errors per word	3.793	0.011	0.036	0.813
	Genre			
Lexical errors per word	5.794	0.000	0.117	0.999
Morphological errors per word	2.086	0.045	0.046	0.797

**Table 7 children-08-01114-t007:** The impact of Group and Genre determinants on error distribution.

Dependent Variables	Independent Variables			
	*F*	*p*	*η* ^2^	*Power*
	Group			
Lexical errors per word	3.849	0.054	0.058	0.489
Morphological errors per word	4.116	0.047	0.062	0.515
Syntactic errors per word	3.520	0.065	0.054	0.455
	Genre			
Lexical errors per word	3.919	0.025	0.112	0.686

## Data Availability

The data presented in this study are available on request from the corresponding author. The data are not publicly available due to original informed consent provisions.

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
