# Peer review of "Lexical and Grammatical Errors in Developmentally Language Disordered and Typically Developed Children: The Impact of Age and Discourse Genre"

_children, 2021, doi:10.3390/children8121114_

Round 1

Reviewer 1 Report

Three main comments.

1. The data obtained in the two studies provide evidence for the relevance of analyzing errors in DLD separately, and the use of different ellicitation methods with 4-6 year old children is a strong component of the paper.

However, the introduction is too long and, occasionally, speculative and vague. The most relevant aspects of the study (motivation for this particular design, justification for the criteria (inclusion/exclusion of participants),  for the typology of errors (why these 4 categories) are missing.

Predictions are not formulated clearly, and consequently the discussion remains descriptive, lacking in depth.

Author Response

Dear Reviewer! Thank you for your valuable comments that helped us to improve our manuscript. We tried to acknowledge all your comments (including the necessity for extensive editing of the English language and style). Please find in the attachments our response with explanations of all the revisions.  manuscript 

Sincerely

The authors of the manuscript     

Reviewer 2 Report

This study examines the impact of age and genre on the types of grammatical and lexical errors in two sets of data. A longitudinal study on children across 4 waves of age and 2 groups of older children -one with typical development and a second one with developmental language disordered. Results show that genre and age affect the type of errors children make.

This is an interesting article that studies a language in need of more research: Russian. However, there are some observations that I consider should be addressed before accepting the manuscript for publication:

Introduction:

  • Some of the citations are too old (1933). There must be new studies that could be used/added.
  • I’m not used to this citation format but I could observe some discrepancies. In some parts of the text the authors use Name [year] and in some others Name[number in the reference list]. Change to the required format.

  • Line 53, include a citation for the statement about children making many errors
  • Lines 59-61, citation for the research supporting that children make errors until the age of 8. Most of the authors report 5 as the age when children have acquired most of the grammatical elements in their language.
  • Line 62. a brief definition for SoT is needed here especially if the authors are making the argument that there has been no attempt to differentiate between SOT and language errors and they have different explanations.
  • Line 84, citation for the scholars researching SOT in adults`
  • Line 88- I don't think the lack of studies is the result of the assumption that language errors and SoT are the result of competence limitations. The authors mention that there are studies about language errors but not SoT.
  • Line 92, citation
  • Page 3, second paragraph, I consider it is important to cite the study conducted by the CATALISE consortium when discussing the term DLD instead of SLI
  • Line 177-181: The authors do not explain the purpose of study 2. It seems that they indirectly talk about it in the paragraph the aim of the study. However, they start with study 2 (with no mention to it) and then include information about study 1. It should be in order: study 1, study 2.

Material and Methods

  • Table 1. did all children complete the 4 waves? No missing data?
  • Line 190, the statement about non-verbal IQ seems out of place, no linked to the text, and not sure if it is part of the table. If it is part of the table, there should be a symbol that directs the reader to the note.
  • Table 2. I suggest deleting the words WORDS in all columns (it is too much text in the table). Instead, the authors can use a note specifying that numbers represent words.
  • Line 198, revise this sentence. Is this a subtitle?
  • Page 5, second paragraph. All this paragraph is confusing. The authors describe the tasks but it is not clear how each task (register, sub-corpus, etc.) was conducted. The use of subtitles may help the reader to identify each of the tasks and their format.
  • Line 215 and 230, what do the authors refer to by “Addressed to adults” and “addressed to peers”? are these terms used instead of Adult-directed speech and Peer-directed speech? If this is the case, please, use the same terms.
  • Line 221, is “Une Kurtinaityte” a person or a company? Word order may give another meaning.
  • Page 6, first paragraph: was “accidentally dropping the images” also used for the fictional tell-peer directed? this part is confusing because the authors mentioned that fictional stories addressed to peers and discussions used the same dolls.

Participants:

  • did you test for language impairment? not being in language therapy does not mean the child has no language disorder, it may be that he/she has not been referred for different reasons.
  • Line 244, same comment as line 190

Procedures:

  • Table 4, what do the numbers represent? the number of words the adult-directed? the average number of words the children used? Also, the same comment for Table 2.
  • Line 253, is task mode the same as register?
  • Line 255, citation, should it be [41]?
  • Line 259, if the goal was to involve the child in a discussion, why not to call this task format discussion as in Study 1?

Section 2.4

  • Line 268, it seems that the word infective does not fit the context.
  • Line 270, is the word GENRE correct in this context? Maybe it is “Gender”?
  • Line 296, please, specify if OTHERS refers to any other type of error not considered in a) to c).
  • 4.2 examples. is this noun masculine in Russian? gloss accordingly and state to what specific type these errors are examples of (gender, tense, etc. ). A person with no knowledge of Russian may not see why this is an error.
  • 4.3 examples: specify if these are errors in agreement, government, incorrect prepositional, etc.
  • Page 8, second paragraph. It may be more informative if you specify what statistical analysis you conducted for each result.

Results:

  • Tables and graphs need an extra row after them. Tables and Figures overlap the text.
  • I suggest that Age should be clearly specified or changed as the variable AGE is not included in any analysis. The authors compare results among waves and it seems to be what the authors use for age. Also, state why you decided those times for each wave.
  • The authors use the variable Genre but in tables 2 and 4 they included register and sub-corpus. Please, use the same terms in tables, analysis, procedures, and results. The reader is not sure if the authors are talking about the same.
  • In the Figures, I suggest setting the Y-axis to the same maximum value. Otherwise, the visual information seems misleading.
  • although it may be obvious, the authors should specify what the bolded numbers mean in the tables.
  • Line 333, add the word quotient before QT as you did in the other cases (Qs, etc.)
  • Line 339, I suggest the authors delete “Descriptive statistics of”. Usually, descriptive statistics are presented in a table, not in a graph.
  • Figure 5, usually interactions are shown with linear graphs so the reader can see where and how the lines cross/how the variables interact
  • From Figure 5 to the last Figure, the authors do not include a mention or reference to them within the text. Please, direct the reader to these Figures.
  • The names of the figures are FIGURE, not Fig as the authors refer to them in the text.

Discussion of Study 1

  • it would be helpful for the reader if the authors restate the purpose or research question for study 1
  • Line 416, do the authors refer to the change observed in each wave? Even though these 2 concepts are related as children were older in each wave AGE per se was not used as a variable.
  • Line 425, this sentence is confusing. Children learn word meaning; that is correct meaning, and learn to differentiate incorrect use but they do not learn the meaning of incorrectly used words.

Results Study 2

  • again, remind the reader about the objective or research question
  • There is no discussion of Study 2. Either include this section or delete Discussion Study 1 and have only one general discussion.
  • Line 484, week does not seem to fit the context, maybe “Weak”
  • Figure 9, move this figure within the results. The discussion section must not have results.

Author Response

Dear Reviewer, 

thank you so much for such a precise and helpful review that enabled us to improve our manuscript. We were trying to acknowledge all the comments and recommendations, correct the typos and vague phrases, and answer all the clarification questions. (Also, and extensive editing of English language and style was done.) Please find in the attachments our detailed response. 

Sincerely,

The authors of the paper
